# The Healthcare and Societal Costs of Familial Intellectual Disability

**DOI:** 10.3390/ijerph21030299

**Published:** 2024-03-04

**Authors:** Deborah Schofield, Rupendra Shrestha, Owen Tan, Katherine Lim, Radhika Rajkumar, Sarah West, Jackie Boyle, Lucinda Murray, Melanie Leffler, Louise Christie, Morgan Rice, Natalie Hart, Jinjing Li, Robert Tanton, Tony Roscioli, Mike Field

**Affiliations:** 1GenIMPACT, Macquarie University, Sydney, NSW 2109, Australia; 2Genetics of Learning Disability (GoLD) Service, Newcastle, NSW 2298, Australia; 3National Centre for Social and Economic Modelling (NATSEM), Institute for Governance and Policy Analysis, University of Canberra, Canberra, ACT 2617, Australia; jinjing.li@canberra.edu.au (J.L.);; 4New South Wales Health Pathology Genomics, Prince of Wales Hospital, Randwick, Sydney, NSW 2031, Australia; 5Centre for Clinical Genetics, Sydney Children’s Hospital, Randwick, Sydney, NSW 2031, Australia; 6Prince of Wales Clinical School, Faculty of Medicine, University of New South Wales, Sydney, NSW 2052, Australia; 7Neuroscience Research Australia, University of New South Wales, Randwick, Sydney, NSW 2031, Australia

**Keywords:** intellectual disability, familial intellectual disability, cost-of-illness, microsimulation, health economics

## Abstract

Most of the studies on the cost of intellectual disability are limited to a healthcare perspective or cohorts composed of individuals where the etiology of the condition is a mixture of genetic and non-genetic factors. When used in policy development, these can impact the decisions made on the optimal allocation of resources. In our study, we have developed a static microsimulation model to estimate the healthcare, societal, and lifetime cost of individuals with familial intellectual disability, an inheritable form of the condition, to families and government. The results from our modeling show that the societal costs outweighed the health costs (approximately 89.2% and 10.8%, respectively). The lifetime cost of familial intellectual disability is approximately AUD 7 million per person and AUD 10.8 million per household. The lifetime costs to families are second to those of the Australian Commonwealth government (AUD 4.2 million and AUD 9.3 million per household, respectively). These findings suggest that familial intellectual disability is a very expensive condition, representing a significant cost to families and government. Understanding the drivers of familial intellectual disability, especially societal, can assist us in the development of policies aimed at improving health outcomes and greater access to social care for affected individuals and their families.

## 1. Introduction

Familial intellectual disability is a term used to describe the occurrence of intellectual disability (ID) among two or more members of a family [1]. In these families, the occurrence of ID, a neurocognitive disorder characterized by significant limitations in cognitive and adaptive functioning appearing and diagnosed before a person is aged 18 years old, is presumed to be caused by the same etiological factors [1]. The severity of familial ID is classified according to the intelligence quotient (IQ) of the affected person, as follows: mild (IQ 50–69), moderate (35–49), severe (20–34), and profound (<20) [2]. The global prevalence of ID is approximately 1%, with an estimated 20–50% of all ID cases of being of genetic etiology, and 450 identified genes are known to cause ID [3]. In Australia, roughly 450,000 (1.8%) of the population was estimated to have ID in 2021 [4].

Familial ID is among the most important unmet diagnostic and management challenges, due to its high prevalence, life-long nature, and rate of recurrence within families [5]. People with familial ID have extensive healthcare and social needs, including that of welfare payments, supported accommodation, equipment such as aids and appliances, and assistance with social activities, such as day programs and leisure activities [5,6]. Those with familial ID often experience comorbidities, such as obesity, feeding issues, physical inactivity, and mental health disorders, significantly impacting their quality of life [6].

However, there are few population-based studies on familial ID, especially on the economic outcomes and demographics [7,8]. Some studies have reported the cost of ID but are limited to the healthcare costs only and may also be limited to the primary diagnosis. Others have reported higher healthcare utilization among those with ID compared to those without ID [9,10]. However, these studies were composed of people with ID of a broad etiology and, thus, were not exclusive to inheritable forms of ID [9,10]. Of the few studies composed of a cohort of people with ID of a genetic origin only, most are on fragile X syndrome (FXS), the most common form of inheritable ID, with annual healthcare costs estimated from USD 2000 to USD 20,000 among adults and children with FXS [11,12,13]. Thus, there is a significant gap in the literature on the cost of ID of genetic etiology, especially for conditions other than FXS, and of costs beyond the healthcare system. Including other costs beyond the healthcare system (using a societal perspective) gives the broader impact of the costs associated with the disorder to the whole society [14]. The societal perspective is particularly important in cost-of-illness studies, as it is often used by policymakers and incorporated into other health economic evaluations for optimal decision making [15]. The societal perspective captures the expenditure on inclusive policies that not only reduce financial inequalities but also other barriers to participation in education, the labor force, and society, factors highlighted by a human rights approach [16]. However, many cost-effectiveness analyses do not use a societal perspective, or the societal perspective is often narrow, with other important costs excluded, for example, special education and transportation costs [17].

We aim to address these gaps in the literature regarding the limited availability of the costs of inheritable forms of ID, such as familial ID. This is achieved by estimating both the healthcare and the societal costs of familial ID and the long-term economic costs to families and government using a static microsimulation model. The findings from our study would benefit other economic evaluations and the development of policies aimed at providing greater access to social support services for those with familial ID and their families. To our knowledge, this is the first study of its kind.

## 2. Materials and Methods

The Genetics of Learning Disability (GoLD) Service serves multigenerational families referred over a 20–30-year period throughout the state of New South Wales (NSW), Australia, creating a unique opportunity to survey costs over different life stages. Primary carers caring for at least one relative with familial ID (ID participant) were recruited from the GoLD Service between 2017 and 2019 to join the Economic and Psychosocial Impacts of Caring for Families Affected by Intellectual Disability (EPIC-ID) study [6]. The carers of the participants with ID who consented to participate in our study were given a questionnaire administered by the EPIC-ID study genetic counsellors through an agreed arrangement of face-to-face interviews or telephone calls occurring shortly after recruitment. The questionnaire collected information about the ID participant’s use of healthcare, supported education, specialized accommodation, respite services, aids and appliances, and home and vehicle modifications. The survey includes questions about co-morbidities, including autism, mental health and behavioral disorders, seizures, hearing loss, and speech difficulties, conditions often associated with ID. We collected information on both the carer and the ID participant’s welfare receipt, income, employment status, assets, and debt. The questionnaire was approved by the Hunter New England Human Research Ethics Committee (NSW HREC Reference No.: HREC/16/HNE/309). Out of the 207 families invited to participate in the EPIC-ID study, 116 agreed to participate and completed the questionnaire, giving a response rate of 56% and a completion rate of 100%. The 116 families comprised 163 ID participants and 105 carers.

A static microsimulation model called IDMOD was developed to assess the cost of caring for families affected by familial ID from multiple perspectives, including the Commonwealth government, and state government, and the families’ healthcare, societal, and opportunity costs, such as income lost, assets lost, and taxes lost. Microsimulation models are a type of mathematical model [18]. Within the microsimulation modeling process, mathematical operations are applied to each microunit (households, individuals or persons, and firms) in the dataset, with every outcome aggregated to form a representative population sample [18]. IDMOD is composed of several unit record datasets, and further details are described elsewhere [6]. The base population of IDMOD is composed of the study participants’ (163 ID participants and 105 carers) demographic information and responses to the survey. For national estimates, the model results were weighted to the Australian population.

The hospital admission and emergency department (ED) admission costs were estimated using the NSW Admitted Patient Data Collection (APDC) and Emergency Department Data Collection (EDDC) linked to IDMOD by the Centre for Health Record Linkage (CHeReL), along with the Australian Refined-Diagnosis Related Group (AR-DRG) codes and the costs obtained from the National Hospital Cost Data Collection. For private hospital patients, we used the Hospital Casemix Protocol average cost estimates for each AR-DRG. The costs of ED presentations were estimated using the Urgency Related Group code derived from the NSW EDDC. The health data linkage was a process where the information from different sources about the same person, family, event, or place were combined into one dataset by CHeReL and provided to the researchers [19]. All patients’ records were successfully matched to the participants in our study through a process to ensure that their personal details were not compromised [19].

We used linked Medicare data in addition to linked hospital data to estimate the cost of medical services, treatments, and medications utilized by the ID participants. Medicare is Australia’s universal health insurance scheme, providing eligible Australians with access to medical services, treatments, and pharmaceutical drugs at a subsidized price [6,20]. The process of creating a linked Medicare dataset involves an individual’s Medicare number and other identifying information [20]. The Medicare number is unique to an individual and must be presented at the time of transaction to receive the subsidy. We used linked Medicare data in addition to linked hospital data to estimate the cost of the medical services, treatments, and medications utilized by the ID participants.

The ID participants were asked to provide information on the use of items not eligible for Medicare, such as over-the-counter (OTC) and complementary medicines, including whether these were publicly funded or an out-of-pocket (OOP) cost. We also collected data on the use and cost of aids and appliances (e.g., wheelchairs, lifting apparatus, hearing aids, and communication aids) and home or vehicle modifications.

The accommodation and care services modeled included supported group accommodation, residential care, and respite care. The use of special education (special school, mainstream school in a support class or with an aide, and early intervention programs) was also captured in the responses, with the cost of these services retrieved from published costs [21]. The OOP expenses were based on the families’ questionnaire responses [6]. The ID participants also reported on the value of their National Disability Insurance Scheme (NDIS) package. The NDIS was established based on international trends of developing individualized funding packages for recipients to access the support services needed [22].

For each person with familial ID, we modeled counterfactual scenarios, such that they had not been affected by familial ID and their carers did not need to care for them, to estimate the income and assets that they could otherwise have achieved and the amount of taxes that would otherwise be paid to the government (income lost, asset lost, and taxes lost, respectively). To estimate the income, assets, and taxes lost, we imputed the income information from STINMOD, a static microsimulation model on Australia’s income and transfer system, on people with otherwise broadly similar characteristics from the general population onto IDMOD’s base population by synthetically matching the participant records with similar sociodemographic characteristics in the two datasets [6,23]. The variables of age, gender, primary state of residence, and highest level of education attained were used to synthetically match the participants from our study to those in STINMOD, as they are associated with income and are unlikely to change among individuals [6,20]. However, because most people with familial ID have a lower level of education than the general population, we generated a counterfactual education variable (the education that they could have achieved if they were not affected by familial ID) for each ID participant using a two-step approach, as follows: (1) we found the proportion of people attaining the education category specified in the 2016 Australian Census and using cumulative percentages to establish the upper and lower bound range of each category [24]; (2) then, we assigned a random value between 0 and 1, drawn from a uniform distribution, to all ID participants. If this random value lay between an upper and lower bound range listed in one of the categories of education attained when we controlled for age, sex, and primary state of residence, this attained education category was assigned to the ID participant as the counterfactual education and was then used along with the other variables in the synthetic matching process [6].

The synthetic matching process was replicated 1000 times using the unrestrictive random sampling method, and the average derived from these 1000 replications was used as the counterfactual income, asset, and tax value [6,20]. The difference between the counterfactual taxes and the current taxes that we estimated our ID participants and carers to have paid to the government denotes the amount of taxes that the government lost from our ID participants and carers, because of their condition or care duties, respectively, influencing their capacity to participate in the workforce [6].

Welfare or transfer payments such as pensions, allowances, supplementary payments, and family payments for the ID participant and their carer (and where relevant, their spouse) were derived from the survey responses.

The average annual costs and the standard errors of the averages were estimated for a range of costs, including healthcare and societal costs and the lost incomes from the Commonwealth government, state government, and ID participants and carers’ perspectives. The lifetime costs for each household with a relative with familial ID were derived by accumulating the average annual cost of each age group of ID participants multiplied by the number of years spent in the age group, up to age 60.

To estimate the aggregate national costs, the data were weighted to reflect the rate of familial ID within the Australian population. The prevalence of familial ID in Australia was derived from published data and the assumption that 20% of all ID is familial based on a study by Partington et al. (2000) [25,26,27,28,29,30]. For sensitivity analysis, we assumed that familial ID was 15% and 25% of all ID. This estimated prevalence was used to derive the weights for the base population of IDMOD.

All costs are presented in 2021 Australian dollars (AUD). All data analyses were performed in SAS 9.4 (SAS Institute, Cary, NC, USA) and Microsoft Excel. The study data were collected and managed using REDCap electronic data capture tools hosted at Macquarie University, NSW, Australia [31,32]. The questionnaire contained standardized instruments accessed under license. Appendix A contains a list of the items surveyed and the costs retrieved from published sources.

## 3. Results

Out of the 163 participants with ID in our study, most of them (*n* = 129, 79.14%) were males, with the largest age group being 5–18 years of age (*n* = 85, 21.47%) with moderate ID (*n* = 73, 44.79%). Out of the 105 carers in our study, most of them were females (*n* = 93, 88.57%), between 31 and 60 years of age (*n* = 81, 77.15%), and partnered (*n* = 79, 75.24%). Most of the ID participants and carers received welfare payments (*n* = 57, 81% and 65, 71%, respectively) (Table 1).

The annual healthcare costs of the ID participants were the greatest in early life, and again in later life, with the government incurring the greatest expenditure across all of the age groups compared to private OOP costs (Table 2). The ID participants aged 0–4 years had the highest per person total average healthcare costs annually (AUD 21,670), while those aged 19–29 had the lowest (AUD 4471) (Table 2). These findings may be due to the extensive diagnostic odyssey and the search for appropriate treatment in early life [33].

### 3.1. Annual Costs

The government also incurred the greatest expenditure on average annual societal costs than private OOP costs for the ID participants aged 5–60 years old (Table 2). In general, the total average annual societal costs of the ID participants increased with age, as different costs are included in the different stages of life of the ID participant, for example, education costs and specialized accommodation costs (Figure 1). As some of these societal costs, particularly those of education and accommodation, are less reliant on carers, the societal costs relating to carers and their spouse decreased over time (Table 3).

Expectedly, the greatest income lost among the ID participants occurred throughout the typical working age of 19–60 years (Table 2). For the carers and their spouse, the income lost was greatest when the ID participants were in the youngest age groups, decreasing to 0 by the time the ID participant was aged 40–60, when the carers themselves moved beyond working age (Table 3).

The total average annual societal costs are considerably higher than the total average annual healthcare costs, comprising approximately 89.2% of the total average annual costs. A breakdown of the government costs (Commonwealth government and state government) can be found in Appendix A.

On average, the annual cost of NDIS packages for those participants in receipt of NDIS with mild ID was AUD 25,257, increasing to AUD 96,452 for those with severe ID (Table 4). The average cost for those with an NDIS package increased by age, with the average cost rising to AUD 112,506 for those aged 40–60 years (Table 4).

### 3.2. Lifetime Costs

We estimated the total lifetime cost of caring for a person with ID to be AUD 6,957,691 per person and AUD 10,800,988 per household (Table 5). The Commonwealth government incurred the highest total lifetime cost (AUD 6,011,629 per person and AUD 9,332,339 per household), with accommodation comprising the largest cost at AUD 3,071,105 per person and AUD 4,767,525 per household. The total lifetime private OOP costs were estimated to be AUD 2,693,437 per person and AUD 4,181,240 per household, higher than the total lifetime cost to state government at AUD 616,787 per person and AUD 957,489 per household (Table 5).

### 3.3. National Costs

In aggregate, the national total annual cost of familial ID is about AUD 5.7 billion based on our survey data. Based on the administrative data estimates, there are approximately 57,995 people in Australia with familial ID, and the national total annual cost of familial ID through NDIS payments is AUD 5.8 billion. The national total lifetime cost of familial ID is estimated to be AUD 403 billion.

In our sensitivity analysis, we assumed that a person with familial ID was expected to live up to the age of 50–70 years old (rather than 60 years, as in the base case). Using these assumptions, the national total lifetime cost of familial ID was estimated to be AUD 334 billion and AUD 473 billion, respectively.

## 4. Discussion

This study has demonstrated the high healthcare and societal costs of familial ID, with societal costs comprising an extraordinary 89.2% of total lifetime costs. Notably, families bear a very significant financial cost, even in Australia where we have a universal health system, welfare payments, and other subsidies, including the NDIS for those with an intellectual disability. Compared to dementia and mental health disorders, the cost of ID is greater, due to the early onset of the disorder, and families are more likely to struggle financially and have a higher caring load [34]. In Australia, the cost of dementia in 2016 was estimated to be AUD 14.25 billion (AUD 35,550 per person per year), with 62% composed of healthcare costs and 38% of societal costs [35].

There are few studies reporting the cost of familial ID or other genetic forms of ID. In one study on the cost of FXS, the total average annual cost of people with FXS from five different European countries ranged from EUR 4951 to EUR 58,862 per person in 2012 [36]. However, these estimates were derived solely from survey responses, with three of the eight countries excluded from the analysis because their sample size consisted of less than 10 participants [36]. Nonetheless, like our study, the findings suggest that societal costs outweigh healthcare costs, prompting a call for policies to incorporate a reduction in the consequences of FXS for those with the condition and their families [36]. The cost may also vary between countries, due to differences in policy and support provided. Australia, for example, introduced the NDIS, which provides extensive support for people with a disability, including connecting families with service providers and funding for therapies, respite care, residential care, and other support, such as assistance with employment. This is in addition to the disability support pension and carer payment and carer allowance, which are welfare payments. Other countries, such as New Zealand, have offered more generous welfare payments for family carers, with the average rate being equivalent to the average rate of formal carers providing care and support services to a person with a disability or the elderly in New Zealand [37], and the UK has moved to integrate health and social care [38]. In Scandinavian countries, children with disabilities are provided with support to assist with their learning, with many municipal schools having established ‘resource schools’ since 2022 for the enrolment of children with certain disabilities, including physical, neuro-psychiatric, and ID [39]. In developing disability policy, countries are increasingly influenced by global organizations, such as WHO, the UN, and OECD, which monitor signatories (governments) to the standards of the United Conventions on the Rights of Persons with Disabilities (UCRPD), who aim to ensure that their policies are based on a framework of human rights [40]. The influence of human rights on disability policies comprises the following three main dimensions: social protection, civil rights, and labor-market integration, resulting in people with disabilities being seen as active citizens with equal entitlements to those of their peers [40]. In Australia, the human rights commission specifically identified the need to reduce the discrimination of people with disabilities in relation to employment [41].

While quantifying the costs borne by patients, their families, and the government, it is important from a policy perspective to recognize that this reflects greater needs [40]. Quantifying the financial cost of illness to families and the government has significant benefits in a policy context, because government tends to respond to high-cost illness and unmet needs. The high cost borne by families was the driver for the establishment of the NDIS [42]. Recognizing the needs of people with a disability and their families, the Australian government established and funded the NDIS in 2013 [43]. The NDIS provides funding to eligible people with a disability to improve their quality of life and support greater independence and access to new skills and jobs. The NDIS provides a wide range of support for families, including treatments such as physiotherapy and speech therapy, respite care, supported accommodation, and aids and appliances to manage a disability. These supports are in addition to welfare payments such as the disability pension and carer payment. The NDIS supports over 500,000 Australians with a disability to access the services and support that they need [44].

Social protection programs play an important role in reducing barriers and supporting the well-being of people with a disability, including familial ID [45]. This is important when viewing disability policies through a human rights lens [40]. Social protection programs, such as the NDIS, that defray additional costs, including healthcare and aids and appliances, and make engagement in education and employment more accessible, are vital to minimize the gaps in living standards and reduce the risk of poverty [45,46]. Taking a human rights approach, with its focus on inclusion, recognizes that investment in programs such as the NDIS may reduce other costs [16]. This is particularly important when environmental barriers result in higher associated costs to the person with the disability. Taking measures that create a more accessible environment, in turn, may reduce the costs to people with a disability and the government. For example, workplace modifications and accessible public transport may reduce private transport costs and increase the rate of employment [45].

Amongst the studies on the cost of ID not specific to familial ID, Arora et al. (2020) [47] estimated the annual cost of ID to be AUD 72,027 per child. However, this was based on a cohort of children aged 2–10 years old, and the healthcare costs included the health services utilized by the caregivers relating to the child’s condition [47]. In a study of 80 Canadian families, the parents whose child had severe ID had higher median annual parental and societal costs (CAD 63,978 and CAD 46,470, respectively) than those parents caring for a child with mild or moderate ID [48]. In summary, these and similar studies have significant limitations for use in studies of targeted interventions and service planning for these groups, in that the cause of ID is unknown, with some including conditions other than ID, and many omitting substantial societal costs and costs in later life [49].

Our study has some limitations, in that the service use relied on self-report, with the use of participants’ recollection of events through surveys over telephone calls and in-person interviews. The use of surveys to collect data also resulted in a small sample size for our study (116 families out of the 207 recruited), due to the demands on the time of families with high levels of care needs. While reduced employment was included in this study, absenteeism was not. Another limitation is that a proportion of the ID participants were interviewed before the NDIS program was fully implemented, and, thus, we did not have the NDIS information for all of the participants. Therefore, the average annual NDIS costs reported in our survey are conservative (thus, we used administrative data for national estimates).

Our study had two particular strengths. Firstly, it was undertaken within a familial clinical genetics service, ensuring that the cohort was well aligned with our study. This setting allowed us to recruit participants with a clinical confirmation of familial ID, excluding those with ID caused by non-genetic factors. As the clinic is a state-wide service, we were able to recruit participants from across NSW, resulting in a broad geographic representation of families from urban, rural, and remote areas. Secondly, we used data linkage to provide robust and comprehensive data on a large proportion of the healthcare costs, such as the hospital admissions and the utilization of healthcare resources listed under Medicare. When available, administrative data provide a reliable source of data compared to study participants’ recollection of events [3]. Data such as that collected in our study are important for capturing the wide-ranging healthcare and societal costs associated with familial ID in order to understand the broad financial impact on families and the government. In our study, taking account of lifetime costs and costs across sectors implicitly captures the downstream benefits of early investments in healthcare and programs such as the NDIS and disability-inclusive education and employment programs [16], while identifying the remaining high private costs to families.

Genomic medicine can improve the health of individuals with familial ID via the provision of a molecular diagnosis, providing access to support services, as well as providing access to targeted therapies and reproductive technologies [50]. Some examples of ways in which genomic medicine can improve the health of individuals with familial ID include therapeutic approaches such as metabolic manipulation to reverse phenotypes, such as enzyme replacement therapy (for example, that used to improve cognition in Hurler’s syndrome, especially if coupled with stem cell therapy); and unexpected off-target effects of existing compounds, e.g., the treatment of MAO-A deficiency in Brunner syndrome with a serotonin reuptake inhibitor and antisense oligonucleotide (ASO) therapies to enhance translation functional protein [51,52]. Other experimental technologies, such as gene therapies, for a range of ID disorders are in phase I or II trials [52]. Genomic medicine also facilitates informed reproductive planning and access to reproductive technologies and, therefore, a reduced rate of recurrence [49]. Such interventions that result in better health outcomes or prevent the recurrence of ID may have a significant impact on the high cost of ID.

Economic evaluations of these promising interventions are important for policies on disabilities; however, it will be important to take a long-term and cross-sector approach in order to ensure that the full benefits of the investment of public funds are captured and cost-effective ways to reduce exclusion while also adopting a human rights lens are recognized.

## 5. Conclusions

The findings from our study suggest that familial intellectual disability is a very expensive condition, with high societal costs compared to healthcare costs. Studies such as ours can serve as inputs for future economic studies, including those on policies designed to improve the affordability and accessibility of health and social support services for those with familial ID. In particular, although the Australian government provides support through education, welfare, and the NDIS policy, the private costs to families remain very high, and there is considerable room to reduce the inequality between families affected by ID and those who are not.

## Figures and Tables

**Figure 1 ijerph-21-00299-f001:**
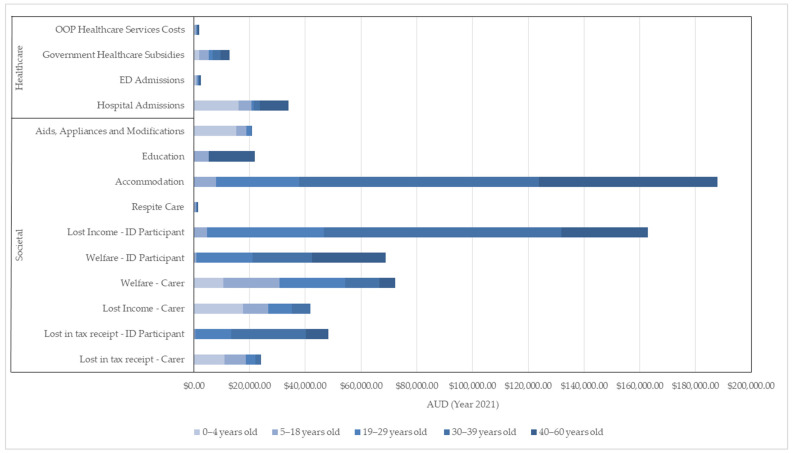
Breakdown of Annual Average Healthcare and Societal Costs by Age Group.

**Table 1 ijerph-21-00299-t001:** Characteristics of the study participants.

	ID Participant	
	Number (*n*)	Percentage (%)
Gender		
Female	34	20.86
Male	129	79.14
Age Group		
0 to 4	9	5.52
5 to 18	85	21.47
19 to 29	35	9.2
30 to 39	15	11.66
40 and above	19	52.15
ID Severity		
Mild	39	23.93
Moderate	73	44.79
Severe	51	31.29
Welfare Payments (>18 years)		
No	13	19
Yes	57	81
	Carers	
	Number (*n*)	Percentage (%)
Gender		
Female	93	88.57
Male	12	11.43
Age Group		
21 to 30	2	1.9
31 to 40	24	22.86
41 to 50	29	27.62
51 to 60	28	26.67
61 to 70	15	14.29
71 or older	7	6.67
Relationship Status		
Partnered	79	75.24
Single	51	31.29
Welfare Payments (>18 years)		
No	26	29
Yes	65	71
Number of ID Participants Cared for by Carer		
Cares for 1 person	53	51.96
Cares for more than 1 person	51	49.04

**Table 2 ijerph-21-00299-t002:** Estimated average annual cost (standard error) related to participants with ID by age group, 2021 AUD.

Age of ID Participant (*n*)	Government (Commonwealth and State) ^1^			Private OOP			Total Annual Average Cost per Person ^4^	Total Annual Average Cost per Household
	Healthcare	Societal ^2^	Taxes Lost	Healthcare	Societal ^3^	Lost Income		
0 to 4 (9)	AUD 21,046	AUD 899	AUD 0	AUD 624	AUD 13,326	AUD 0	AUD 35,894	AUD 55,721
	(AUD 12,154)	(AUD 842)	(AUD 0)	(AUD 236)	(AUD 9562)	(AUD 0)	(AUD 16,130)	(AUD 25,040)
5 to 18 (85)	AUD 10,788	AUD 51,623	AUD 627	AUD 1034	AUD 2131	AUD 5134	AUD 69,574	AUD 108,005
	(AUD 1664)	(AUD 5618)	(AUD 142)	(AUD 362)	(AUD 554)	(AUD 998)	(AUD 5973)	(AUD 9272)
19 to 29 (35)	AUD 4113	AUD 59,410	AUD 14,076	AUD 358	AUD 1060	AUD 44,605	AUD 87,654	AUD 136,072
	(AUD 772)	(AUD 11,498)	(AUD 1236)	(AUD 125)	(AUD 249)	(AUD 3161)	(AUD 11,616)	(AUD 18,032)
30 to 39 (15)	AUD 4807	AUD 115,540	AUD 28,647	AUD 227	AUD 377	AUD 91,248	AUD 189,783	AUD 294,616
	(AUD 849)	(AUD 34,230)	(AUD 1229)	(AUD 100)	(AUD 218)	(AUD 3823)	(AUD 33,573)	(AUD 52,118)
40 to 60 (19)	AUD 14,416	AUD 99,143	AUD 9638	AUD 738	AUD 1295	AUD 31,482	AUD 119,489	AUD 185,492
	(AUD 7828)	(AUD 20,275)	(AUD 1926)	(AUD 218)	(AUD 1098)	(AUD 8783)	(AUD 27,178)	(AUD 42,191)

Notes: ^1^ A breakdown of the Commonwealth and state governments’ costs is provided in Appendix A. ^2^ Comprises welfare payments, special education and disability supports (including NDIS support accommodation, aids, appliances, and modifications to house or car), costs for Commonwealth government and accommodation supports and special education costs for state government. ^3^ Expenses for aids, home and vehicle modification, accommodation, and supplements/non-prescription medicines/special diets. ^4^ Welfare payments and taxes lost were excluded to avoid double counting.

**Table 3 ijerph-21-00299-t003:** Estimated average annual cost (standard error) related to carers and their spouse by ID participants’ age group, 2021 AUD.

Age of ID Participant (*n*)	Government (Commonwealth and State) ^1^		Private OOP	Carer and Spouse Total Average Annual Cost ^3^	Total Annual Average Cost per Person (Including ID Participant, Carer, and Spouse)	Total Annual Average Cost per Household (Including ID Participant, Carer, and Spouse)
	Societal ^2^	Taxes Lost	Carer and Spouse Lost Income			
0 to 4 (9)	AUD 10,775	AUD 7190	AUD 15,387	AUD 15,387	AUD 51,281	AUD 79,608
	(AUD 4368)	(AUD 8284)	(AUD 6346)	(AUD 6346)	(AUD 18,785)	(AUD 29,161)
5 to 18 (85)	AUD 10,002	AUD 8556	AUD 10,295	AUD 11,293	AUD 80,867	AUD 125,536
	(AUD 1092)	(AUD 1545)	(AUD 1806)	(AUD 1989)	(AUD 6222)	(AUD 9659)
19 to 29 (35)	AUD 9059	AUD 3600	AUD 8352	AUD 10,311	AUD 97,965	AUD 152,080
	(AUD 1925)	(AUD 1530)	(AUD 2167)	(AUD 2249)	(AUD 11,255)	(AUD 17,472)
30 to 39 (15)	AUD 6250	AUD 1836	AUD 6594	AUD 8445	AUD 198,228	AUD 307,725
	(AUD 2224)	(AUD 1583)	(AUD 2064)	(AUD 2809)	(AUD 33,833)	(AUD 52,522)
40 to 60 (19)	AUD 6087	AUD 380	AUD 0	AUD 0	AUD 119,489	AUD 185,492
	(AUD 1806)	(AUD 2854)	(AUD 0)	(AUD 0)	(AUD 27,178)	(AUD 42,191)

Notes: ^1^ A breakdown of the Commonwealth and state governments’ costs is provided in Appendix A. ^2^ Comprises welfare payments and housing support costs. ^3^ Welfare payments and taxes lost were excluded to avoid double counting.

**Table 4 ijerph-21-00299-t004:** Summary statistics and annual average NDIS package cost per NDIS recipient.

	Number of NDIS Recipients	Mean	Standard Error
ID Severity			
Mild	13	AUD 25,257	AUD 20,710
Moderate	30	AUD 56,453	AUD 77,212
Severe	28	AUD 96,452	AUD 83,795
Age Group			
0 to 4	0	AUD 0	AUD 0
5 to 18	43	AUD 44,538	AUD 65,135
19 to 29	21	AUD 77,104	AUD 60,414
30 to 39	3	AUD 65,835	AUD 42,767
40 to 60	11	AUD 112,506	AUD 112,256

**Table 5 ijerph-21-00299-t005:** Lifetime costs of ID participants, 0–60 years old, 2021 AUD.

	Per Person	Per Household
Commonwealth Government		
Health system	AUD 432,444	AUD 671,318
Education	AUD 110,404	AUD 171,389
Accommodation	AUD 3,071,105	AUD 4,767,525
Welfare	AUD 1,060,178	AUD 1,645,800
Subsidy for aids, appliances, and modifications	AUD 33,514	AUD 52,026
Lost in tax receipt due to ID participant	AUD 652,484	AUD 1,012,904
Lost in tax receipt due to carer and spouse	AUD 221,674	AUD 344,122
Total Commonwealth Government costs	AUD 6,011,629	AUD 9,332,339
State Government		
Health system	AUD 219,872	AUD 341,326
Education	AUD 287,658	AUD 446,555
Accommodation	AUD 55,225	AUD 85,730
Welfare (public housing)	AUD 54,032	AUD 83,878
Total State Government costs	AUD 616,787	AUD 957,489
Private OOP		
Health system	AUD 39,304	AUD 61,015
Accommodation	AUD 28,217	AUD 43,804
Aids, appliances, and modifications	AUD 110,886	AUD 172,137
ID participant’s lost income	AUD 2,136,146	AUD 3,316,112
Carer and spouse lost income	AUD 378,884	AUD 588,172
Total private OOP costs	AUD 2,693,437	AUD 4,181,240
Total cost (after adjusting for welfare compensating lost income)	AUD 6,957,691	AUD 10,800,988

## Data Availability

The data are not publicly available due to patient confidentiality and privacy.

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
