# Peer review of "The Healthcare and Societal Costs of Familial Intellectual Disability"

_ijerph, 2024, doi:10.3390/ijerph21030299_

Round 1
Reviewer 1 Report
Comments and Suggestions for Authors
This study is quite important, and it is a well done project. Here are some minor suggestions for the authors to improve their manuscript.
1.Please delete the extra words in the keywords (for example, 1, 2, 3, 4, and 5); and list three to ten pertinent keywords specific to the article yet reasonably common with the subject discipline.
2.Please delete the extra space in lines 74 (page 2), 184 (page 4), 190-193 (page 4), 198-199, and 207 (page 5).
3.Please put table 2 on page 6.
4.Figure 1 is not clear and can be made much more beautiful.
5.Please add the policy suggestions for other countries as well.
6.Reference style should be checked.
Reviewer 2 Report
Comments and Suggestions for Authors
This paper is an analysis of the healthcare and societal costs of familial intellectual disability (ID). The angle of approach to familial ID is narrow and purely medical, which is very problematic as it focuses on familial ID as a societal burden.
While the study initially mentions the need to estimates the benefits and the costs of the impairment (“Including other costs beyond the healthcare system (using a societal perspective) gives a broader impact of the cost and benefits associated with the disorder to the whole society”), the study ignores benefits entirely and estimates costs only.
As a result of such a narrow frame, the implications derived at the end of the paper are only about preventing familial ID. Yet prevention will never be entirely effective and familial ID will persist. Persons with ID have human rights (Robinson and Fisher 2023). Perhaps the services under consideration are not sufficient or of adequate quality for the human rights of persons in familial ID to be fulfilled.
A potential unintended effect of such a study is that readers could take away from this analysis the need to cut the policies and programs that families with ID depend on given their costs to societies. When economic approaches such as the one used in this paper are divorced from a human rights perspective, they can lead to unintended consequences that undermine the progression to a fully inclusive society.
While the benefits of familial ID are not estimated in this paper and are in general hard to quantify, they need to be discussed.
The economic lens used in this paper should be wide enough to incorporate the fundamental notion of human rights for families with ID (Robinson and Fisher 2023), which may be very challenging to do. At the very least, the introduction and discussion/conclusion should be rewritten to this effect.
An alternative would be to focus on the costs born by families with ID, which would not have the risk of the unintended effect mentioned above.
Robinson, S. and Fisher, K. (eds). Research Handbook in Disability Policy. Elgar. https://www.e-elgar.com/shop/usd/research-handbook-on-disability-policy-9781800373648.html
Comments on the Quality of English LanguageEnglish language is good. Some edits are needed here and there, including in the abstract.
Reviewer 3 Report
Comments and Suggestions for Authors
Through a static microsimulation model, the authors estimated the healthcare and societal cost, and lifetime cost of individuals with familial intellectual disability, to families and governments. The authors used wide range of data sources including questionnaire to estimate health care costs, societal costs and life time cost of ID participants.
The manuscript is well articulated and presented in scholarly manner. The methodology section is comprehensive, and described the process involved in estimating these costs. However, the following are some of my comments
1. In introduction section authors may provide certain statistical information about the ID in the country (economic burden, prevalence rate etc.) and facilities available for treatment at country level.
2. In methodology section- data collection through questionnaire should also include the time frame for conducting interview, how recall bias or systematic error that occurs when participants do not remember previous events or experiences etc., to be addressed.
3. In methodology section, it may be wise to give the details of costs included/collected for estimating healthcare cost, societal cost and life time costs in a tabular form including source of data.
4. While calculating income loss, it’s not clear whether absenteeism from work is included?
5. It’s not clear whether the modeling considers discounting techniques while estimating life time costs
6. Are there issues related to comorbidities in ID patients? If so, how was it controlled in estimating health care costs?
7. Conclusion section needs to be elaborated by including policy implications on the government.
Round 2
Reviewer 2 Report
Comments and Suggestions for Authors
The introduction and the discussion still reflect a narrow and primarily medical understanding of the issue at stake.
The authors now state: “While recognizing the cost burden for patients, their families and government, it is important from a policy perspective, that it should not be interpreted as suggesting that people suffering disability or illness have less value (39), but rather, that they have greater needs.”
Some of the costs borne by households and governments are not the result of ‘greater needs’ but of environmental barriers. For instance, if the physical environment (e.g. sidewalks, buses) has physical barriers then persons with disabilities will spend more money on private transportation. Discriminatory attitudes in the labor market makes it harder to get jobs and therefore makes people more likely to apply for social benefits to make ends meet. Therefore, it is problematic to present such costs as a burden. They are a burden indeed on families and governments, but it is not the ‘greater needs’ that are burdensome, it is an inaccessible society, at least in part. This should be conveyed and the study should be more nuanced than a burden study.
In addition, the authors need to engage more with Robinson & Fisher (2023), which has many contributions on the economics of disability rights. In particular, the introductory chapter by Mont notes the limitations of costs efficiency in the disability field, in particular the risk of going for ‘low hanging fruits’ when it comes to reducing costs to the detriment of human rights. Mont also notes that the need to identify cost-effective ways to reduce exclusion while also adopting a human rights lens.
A human rights lens should be added to this study and how it is framed for it not to have detrimental effects and for it not to reinforce a view of people with familial ID that is focused on needs and cost burdens.
